# Exploring Anastomosis of Hyphae and Mating-Type Compatibility of *Pochonia chlamydosporia* Isolates of the *Meloidogyne*, *Heterodera* and *Globodera* Biotypes

**DOI:** 10.3390/pathogens11060619

**Published:** 2022-05-26

**Authors:** Mariella Matilde Finetti-Sialer, Rosa Helena Manzanilla-López

**Affiliations:** 1Istituto di Bioscienze e Biorisorse, Consiglio Nazionale delle Ricerche, Via G. Amendola 165/A, 70126 Bari, Italy; 2Centro de Desarrollo de Productos Bióticos, Instituto Politécnico Nacional, Carretera Yautepec-Jojutla Km 85, San Isidro 62739, Mexico

**Keywords:** *Pochonia* biotypes, *MAT1-1-1*, *MAT1-2-1*, biological control agents, anastomosis

## Abstract

The endophytic and nematophagous fungus *Pochonia chlamydosporia* is an efficient biological control agent of plant-parasitic nematodes. Isolates of the fungus can be allocated to a biotype group according to the nematode host, but it is unknown if genetic interchange can occur between different biotypes, which may affect their parasitic performance. An anastomosis assay was conducted in vitro to assess hyphae vegetative compatibility/incompatibility followed by a PCR-based mating-type assay genotyping of five isolates of *P. chlamydosporia* var. *chlamydoporia* of the *Meloidogyne* sp. (Pc10, Pc190, Pc309), *Globodera* sp. (Pc280) and *Heterodera avenae* (Pc60) biotypes, including 16 pairwise isolates combinations in four replicates. Pairwise combinations were tested on glass slides and mycelia were stained to confirm nuclei migration between anastomosing hyphae using fluorescence microscopy. Anastomosis only occurred between mycelium hyphae of the same isolate and biotype. Mating-type PCR-based molecular assays showed that all isolates were heterothallic. The *MAT1-1* genotype was found in isolates Pc10, Pc190, Pc280, Pc309, and the *MAT1-2* genotype in Pc60. The results showed a vegetative incompatibility among isolates, suggesting the occurrence of such interactions for their respective biotypes. Anastomosis and PCR mating-type results suggest that different fungal biotypes can occur in the same niche but that genetic incompatibility mechanisms, such as mating-type, may limit or impede viable heterokaryosis.

## 1. Introduction

Plant-parasitic nematodes (PPN) cause annual yield losses estimated in the order of USD 12.5 billion, worldwide [1]. Management of PPN is usually carried out through an Integrated Pest Management (IPM) approach, which can include application of PPN biological control agents (BCA). Among the clavicipitaceous nematophagous fungi, *Pochonia chlamydosporia* var. *chlamydosporia* is characterized by diversity of econutritional groups [2,3], acting as a soil saprotroph, an endophyte, an antagonist of phytopathogenic fungi [4,5], and a parasite of beetle larvae, mollusk eggs and PPN [6]. *Pochonia chlamydosporia* is also considered a successful BCA of PPN [7]. Isolates of the fungus can be assigned to a biotype according to the nematode host from which isolates were originally isolated. Isolates show a nematode host preference (biotype) but not host specificity, thus being able to parasitize more than one host. The fungus is a facultative parasite of eggs and sedentary females of root-knot (*Meloidogyne* spp.) [8], false root-knot (*Nacobbus* spp.) [9] and cyst nematodes (*Globodera* spp., *Heterodera* spp.) [10]. In addition, endophytic effects of *P. chlamydosporia* [11] have been linked to the control of endoparasitic migratory nematodes, such as *Pratylenchus goodeyi* [12] and the burrowing nematode *Radopholus similis*, as well as to the activation of host plant defence mechanisms [11,12].

Application of a single biotype of the fungus as a biological control agent vs. a specific host nematode species may represent a successful strategy, however it requires careful isolate selection [7,13] due to other important characteristics. Such factors include the isolates’ performance as good or poor soil colonizers, rhizosphere and nematode egg parasites, which in turn can be affected through interaction or competition with other indigenous isolates and biotypes [14,15,16]. For example, the performance of isolate Pc-10 (RKN biotype) applied to control *M. javanica* was different when Pc-10 was applied alone or in a mixture of isolates (Pc-10, Pc-3, Pc-28) [17]. Although *M. javanica* egg numbers decreased in all combinations, Pc-10 biocontrol did not increase when mixed with any other of the two isolates. Among possible explanations, an apparent lack of synergism between the isolates to increase nematode control was considered, due to isolate incompatibility. However, such biological factors, including genetic variation and compatibility, remain to be explored for this BCA.

Host-related genetic variation occurs at the infraspecific level in *P. chlamydosporia* isolates [18]. Whether isolates of different biotypes and genotype profiles are vegetative-compatible and capable of asexual genetic interchange is yet unknown. Such compatibility may affect *P. chlamydosporia* efficacy as a BCA, including the stability of strains, i.e., a group of clonally related individuals [19], as well as their virulence and parasitism effectiveness. The fungus stability appears as a relevant issue, because the asexual interchange of nuclei can result in heterokaryosis, i.e., two or more nuclei of different lineages, coexisting in the same hyphae and mycelia cytoplasm. Such a condition may possibly add or alter genetic capabilities through a parasexual process, in addition to the normal sexual cycle, as shown in many fungi [20,21]. Mating fusion events taking place in the sexual cycle depend on genotypic cell types under the control of mating-type genes [22]. Mating-types MAT1 and MAT2 are determined by a single genetic locus, *MAT1*, and the respective alleles or idiomorphs *MAT1-1* and *MAT1-2* [20,22]. Heterothallic fungal strains have one of the two alleles, whilst homothallic strains have both.

The PCR-based mating-type assay developed by Yokoyama et al. [3,23,24] for *Clavicipitaceae* species provides a fast and practical tool to identify the mating-type genotype of *P. chlamydosporia* strains or biotypes, without the need for mating experiments. Yokoyama et al. [3] primer sets were designed for identifying the mating-type genes *MAT1-1-1* and *MAT1-2-1* based on the amino acid sequences of the conserved alpha and HMG boxes [3,22]. Furthermore, *MAT1-1-1* and *MAT1-2-1* (expressed in vegetative mycelia) can be involved in vegetative incompatibility [22,23]. Mating-type for most *P. chlamydosporia* isolates and strains remains unknown thus far, except for a couple of reports by Yokoyama et al. [3,24]. More research on the mating-type of *Pochonia* species and varieties will help to assess the homothallic or heterothallic nature of isolates, strains and biotypes, as well as potential vegetative compatibility/incompatibility. The objectives of the present study were: (i) to assess, through an in vitro anastomosis assay, the vegetative compatibility of five *P. chlamydosporia* var. *chlamydosporia* isolates belonging to the root-knot (*Meloidogyne*) or cyst nematode biotype (*Heterodera*, *Globodera*); and (ii) to identify, through a molecular approach, their *MAT1-1* or *MAT1-2* mating-type genotype.

## 2. Results

The fluorescent dye HOE 33258 [25] allowed for observation of hyphal cell walls, septa, hyphal pegs, conidia and hyphal nuclei (Figure 1). Hyphal pegs and hyphal bridges formed between anastomosing hyphae (Figure 2A arrows). However, hyphal pegs were not always conspicuous, e.g., Pc190 × Pc280, Pc60 × Pc309 (Figure 2B). A total of 68 anastomosing hyphae pairs were examined for nuclei migration, which appeared to be a common event within mycelium hyphae of all individual colonies of isolates (Figure 1B). Anastomosis accompanied by nuclei migration between hyphae of two different colonies of the same isolate was also observed (7%) in pairwise combinations Pc60 × Pc60, Pc190 × Pc190, Pc280 × Pc280, and Pc309 × Pc309 (*n* = 5). Nuclei migration was not observed at anastomosing points between hyphae of different isolates of the RKN, CCN or PCN biotype in pairwise combinations Pc190 × Pc60 (*n* = 6), Pc190 × Pc280 (*n* = 9), Pc60 × Pc280 (*n* = 1). Migrating nuclei (Figure 2C) between anastomosing hyphae of different isolates and biotypes, i.e., Pc280 × Pc309 (*n* = 5), could not be confirmed using a 63× immersion objective. Hyphal death was also observed at contact points between different isolates in pairwise combinations Pc309 × Pc60, Pc280 × Pc190.

The mating-type molecular assay (Table 1) revealed the heterothallic nature of all five isolates (including Pc10, positive control) that belonged either to the *MAT1-1-1* or *MAT1-2-1* genotype. PCR amplification of the mating-type genes of isolates and respective biotype was obtained with two of the four primer pairs tested (Table 2). *MAT1-1-1* gave an amplicon of 230 bp (including primers), that was visualised in isolates Pc10, Pc190, Pc280 and Pc309, whereas using *MAT1-2-1* only, isolate Pc60 gave a clear band of ca. 250 bp (Figure 3). The molecular results agreed with the anastomosis assay observations made with light and fluorescence microscopy and suggested isolate incompatibility.

## 3. Discussion

According to Krnjaja et al. [26], vegetative or heterokaryon compatibility or incompatibility tests refer to ‘visual assessment of phenotypes’ of two strains of one fungal species cultivated in a mixed culture on a specific medium, which enables identification of fungal clones. Hyphae of different isolates that are vegetatively compatible can anastomose, fuse and exchange cytoplasmic or nuclear content to form a viable heterokaryon. Vegetatively incompatible isolates cannot anastomose [26]. Fluorescence microscopy observations showed nuclei migration between anastomosing hyphae of each of the four *P. chlamydosporia* var. *chlamydosporia* isolates when stained with HOE 33258. This type of anastomosis contributes to an interconnected hyphal network allowing for nutrient distribution and signals that spread within the mycelium colony via cytoplasmic flow [27].

Anastomosis also occurred between combinations of the same fungal isolate and biotype (i.e., CCN Pc60 × Pc60, PCN Pc280 × Pc280, and RKN Pc190 × Pc190, Pc309 × Pc309). Although hyphae contacts between the same or different isolates and biotypes was observed in the other pairwise combinations, i.e., Pc190 × Pc309 (RKN-RKN) and Pc60 × Pc280 (CCN-PCN), it was not possible to confirm nuclei migration, and hence vegetative compatibility.

The vegetative compatibility/incompatibility of the four isolates of *Pochonia* was further explored with Yokoyama’s et al. [3] PCR-based mating-type assay for *Clavicipitaceae*. Results showed that the five isolates (i.e., including the positive control Pc10) of *P. chlamydosporia* var. *chlamydosporia* are heterothallic, which may help partly to explain the vegetative incompatibility observed in the anastomosis assay. Four isolates showed the *MAT1-1* genotype, i.e., RKN (Pc190, Pc309, Pc10) and PCN (Pc280) biotype, whereas the *MAT1-2* genotype was only found in the CCN (Pc60) biotype.

It has been reported that both mating-type genotypes can occur in different eco-nutritional groups of *Clavicipitaceae*, including *P. chlamydosporia* [3,24]. Based on PCR mating-type assay *MAT1-1-1* and *MAT1-2-1* primers, the *MAT1-1* genotype has been previously reported [3] for *Cordyceps chlamydosporia* (=*P. chlamydosporia*) strain NBRC 9249 (*MAT1-1-1)* and *MAT1-2* genotype for strain IAM 14700 (*MAT1-2-1*) [24], both derived from a soil source, and *MAT1-1* from *P. suchlasporia* var. *suchlasporia* (=*Metapochonia suchlasporia*) strain IAM 14707 (*MAT1-1-1*) [24], proceeding from an unknown nematode species source. The *MAT1-2* genotype was only reported for *P. suchlasporia* var. *catenata* (= *M. suchlasporia*) strain NBCR 32276 (*MAT1-2-1*), from a soil source [24]. Since these reports were published, the taxonomy of *Pochonia* has changed, *P. suchlasporia* being moved to *Metapochonia, ‘a genus with a phylogenetic position near to but outside the true Pochonia clade’* [6]. Our results confirmed the presence of genotype *MAT1-1* (*MAT1-1-1*) in *P. chlamydosporia* var. *chlamydosporia* isolates that belonged either to the RKN (*Meloidogyne* spp.) or PCN (*G. rostochiensis*) biotypes, and for the first time, we report the *MAT1-2* (*MAT1-2-1*) genotype in one isolate (Pc60) of the CCN *H. avenae* biotype. Research on the MAT1 genotypes of different *P. chlamydosporia* varieties and strains may help not only to investigate their homothallic or heterothallic nature, but may contribute to eco-nutritional and phylogenetic analyses [3,24].

Mating-type could also be useful to investigate asexual and sexual reproductive patterns, including perithecium formation [23], of *P. chlamydosporia* and anamorph-teleomorph connections [6,28,29]. Molecular mating-type investigations helped to reveal the existence of species populations that lack obvious sexual stages in nature but that can harbour patterns of sexuality [29]. Self-fertile homothallic species do not require a mating partner to complete sexual reproduction, but heterothallic fungi are self-sterile and only mate with a haploid cell of the opposite mating-type [30]. Heterothallic genetic barriers and sexual identity are established by strain mating-type genes, which encode transcriptional regulators to control the expression of many genes for sexual compatibility and reproduction. These include the mating-type-specific pheromone and its G-protein-linked receptor [30], which have not yet been investigated for *Pochonia*.

Other important factors that need to be considered when investigating vegetative compatibility/incompatibility of *P. chlamydosporia* include host preference, competition, geographic origin and genetic profile. Some isolates are able to parasitize both RKN and CN eggs [7], but egg parasitism in a biotype tends to be higher on its ‘preferred’ host, i.e., the one from which it was originally isolated. Morton et al. [31] found that *P. chlamydosporia* isolates from soils infested with *Heterodera* spp. formed a different group to those infested with *Meloidogyne* spp., thereby suggesting that genetic variation and adaptation relate to the host. The capacity to infect RKN or CN eggs has been linked to amino acid sequence polymorphism of the VCP1 serine protease [32]. However, for some isolates, the *vcp1* gene polymorphism of isolates from CN or RKN do not always match the original nematode host or biotype reported [7].

Isolates host preference and fungal virulence can also be investigated using MAT-based phylogenetic data [33]. It has been proposed that host switching and virulence are acquired traits related to diverse sexual life histories and speciation in the *Clavicipitaceae* genera *Metarhizium* and *Pochonia*. These comprise a monophyletic clade of fungi in a long-term association with plants and a pathogenic association with nematodes or insects [34].

In conclusion, compatibility between *P. chlamydosporia* isolates and respective biotype is an important and complex issue regarding potential interactions between native isolates (and isolates or strains) introduced for nematode biocontrol. Our preliminary results on anastomosis and PCR mating-type support the hypothesis that different fungal biotypes can occur in the same niche but that genetic incompatibility mechanisms, such as mating-type, may limit or impede viable heterokaryosis. However, larger samples of isolates and biotypes of *P. chlamydosporia* varieties need to be assessed for vegetative compatibility or incompatibility, as well as *MAT1-1* and *MAT1-2* genotyping and identification of genes responsible for heterokaryon incompatibility. A combination of novel molecular and microscopy approaches, such as isolates transformed with green fluorescent protein [35], confocal fluorescence microscopy and fluorescence life-cell imaging, could be also required to further investigate strain compatibility.

## 4. Materials and Methods

### 4.1. Isolates

The four *P. chlamydosporia* var. *chlamydosporia* isolates (i.e., Pc60, Pc190, Pc280, Pc390) were included in the in vitro anastomosis study and the mating-type molecular assay (Table 3). The four isolates were selected considering their inclusion in previous *P. chlamydosporia* studies related to population diversity and host preference relevance for regulation of nematode populations [7,14]. Two isolates (Pc190, Pc309) were of the RKN biotype *Meloidogyne* sp., one (Pc280) was of the potato cyst nematode biotype (PPC) *Globodera* sp., and one (Pc60) was of the cereal cyst nematode biotype (CCN) *H. avenae*. All four isolates were identified as *P. chlamydosporia* var. *chlamydosporia* [7] and had different geographical origins, contrasting rhizosphere and egg parasitism profiles and DNA fingerprints [7,14,15]. Isolates profiles were as follows: Pc309 (Zimbabwe)—good root coloniser/good parasite, Pc60 (UK)—poor root coloniser/poor parasite, Pc190 (Kenya) and Pc280 (Jersey)—poor root colonizer/good parasite. The same four *Pochonia* isolates, plus one isolate of *Purpureocillium lilacinum* (*Ophiocordicypitaceae*), were sourced from the *Pochonia chlamydosporia* collection of Rothamsted Research Ltd. (Harpenden, UK); a fifth isolate of *P. chlamydosporia*, i.e., Pc10 (Brazil, *Meloidogyne* biotype, isolated by B. R. Kerry), was sourced from the Institute for Sustainable Plant Protection (Bari, Italy). Both Pc10 and *P. lilacinum* were used only as standard positive controls for the mating-type molecular assay [3].

### 4.2. Monosporic Cultures and Medium-Coated Glass Slides

All laboratory assays were made in a flow cabinet under sterile conditions. Monosporic culture of isolates was produced either through conidia dilutions [36,37] or conidia picked individually with a Singer stage-mounted micromanipulator [38]. Monosporic cultures were subsequently cultured on fresh CMA and PDA media with antibiotics [38,39].

Monosporic cultures were first assessed for mycelium growth in CMA, PDA, glucose water agar (0.5% glucose) or Czapek Dox Broth-Agar media with antibiotics for seven days. Based on colony diameter measurements (data not shown), the Czapek Dox Broth-Agar medium (CDBA) was modified to obtain a more synchronous growth of isolates for the anastomosis assay. The modified medium was made of 200 mL of CDBA, i.e., 8.35 g Czapek Dox Broth (Sigma-Aldrich, St. Louis, MO, USA), 3 g Daishin agar (Brunschwig chemie, Amsterdam, the Netherlands), supplemented with 0.5 g glucose (Sigma-Aldrich, St. Louis, MO, USA), 0.1 g yeast (Merck Darmstadt, Darmstadt, Germany), and 0.2 g PDB (potato dextrose broth, Formedium™, Hunstanton, UK) [39,40,41]. Antibiotics included 12.5 mg of chlortetracycline hydrochloride (Sigma-Aldrich), 12.5 mg of streptomycin sulphate (Sigma-Aldrich), 12.5 mg of chloramphenicol, dissolved in 50 mL of sterile distilled water (sdw) added to 200 mL of warm, autoclaved CDBA. To prepare CDBA-coated glass slides (CDBA-CS), one surface of a pre-washed sterile glass slide was coated with a film of warm, autoclaved CDBA [25] and left to solidify. Slides were placed into a sterile Kopling staining jar, sealed with a lid and parafilm^®^ (Sigma-Aldrich, St. Louis, MO, USA), and kept at 4 °C until use.

### 4.3. Anastomosis In Vitro Assay

The anastomosis in vitro assay included 16 pairwise combinations for the four isolates (4 × 4), repeated four times and repeated twice overtime. Each pairwise combination consisted of two PDA-colonized discs (3 mm diameter each) that were placed 0.3–0.5 cm apart from each other onto individual CDBA-CS. Each CDBA-CS with PDA-colonized discs was placed onto a sterile square glass (3.5 cm × 3.5 cm) kept in a Petri dish lined with moistened filter paper. The Petri dish was placed inside a plastic box (17.5 cm × 6.0 cm × 11.5 cm) lined with moistened blue roll and kept in an incubator for five days at 26 °C (Figure 4).

Mycelia of isolates grown onto CDBA-CS were assessed every 24 h for five days with an Olympus inverted microscope (40× objective). PDA-colonised disc pairs were removed once the hyphal tips from both isolates reached each other (Figure 5A,B) [37]. Mycelia on slides were left to dry in a flow cabinet for 10–15 min, and then stained with 40 μL of the HOE 33258 working solution made from 1.2 mL of stock solution (10 mg dye/25 mL of sdw) added to 50 mL of 0.025 M H_3_BO_3_ buffer and 0.1 M NaOH, pH 10.5 [25]. Subsequently, a coverslip (24 mm × 50 mm) was placed on top of the stained mycelia and the CDBA-CS was kept for 20 min in the dark in a sterile Petri dish (10 cm diameter) before being examined with 20×, 40× and 63× objectives in a fluorescence microscope (Zeiss Axioscope, Jena, Germany), with a high-pressure mercury arc lamp as light source. Micrographs were taken with a QImaging camera and MetaMorph ver. 7.6 software (Molecular Devices, CA, USA). Anastomosis was confirmed and scored when nuclei were observed migrating from one hypha to another at anastomosing points.

### 4.4. Molecular Mating-Type Assay

Genomic DNA was extracted from pure cultures of isolates growing actively on PDA. About 50 to 100 mg of mycelium was collected and used for DNA extraction. In brief, the fungus mycelium was scraped and transferred to a 1.5 mL microcentrifuge tube. The fungal tissue was broken down by adding 500 µL of lysis extraction buffer (100 mM Tris HCl, pH: 8, 20 mM EDTA, 1.4 M NaCl) and glass beads/broken cover slips, and vortexed for 2 min (three times interspersed by chilling at −80 °C for 10 min each). The procedure was followed by an incubation at 65 °C for 30 min and chloroform: isoamyl alcohol (24:1) purification. Finally, the nucleic acids were precipitated with isopropanol. Extracted DNA was dissolved in 60 µL of water and frozen in 100 ng/µL aliquots, to be used as template for PCR assays.

For the PCR-based mating-type assay, primers (Table 1) were synthesized according to Yokoyama et al. [3]. DNA diluted to 100 ng/µL was used for the PCR assay. The reactions were carried out in a 25 μL mixture with 10 mM Tris-HCl (pH 8.8; 50 mM KCl; 100 μM each of dATP, dCTP, dGTP, and dTTP; 1.5 mM MgCl_2_; 1.25 units of Taq polymerase; and 500 nM of each primer). The mixtures were incubated in a thermal cycler following the same parameters as described [3]. *Pochonia chlamydosporia* var. *chlamydosporia* isolate (Pc10) and *Purpureocillium lilacinum* were used as positive and negative controls.

Amplified fragments were analysed by electrophoresis in 1.2% agarose gels in Tris-borate-EDTA. The fragments obtained were eluted from the gel and cloned in a dTTP-tailed pGEMT vector according to the manufacturer’s (Promega, Madison, WI, USA) instructions. The recombinant plasmids were used to transform competent cells of *Escherichia coli* isolate DH5α. DNA from recombinant plasmids were prepared and the insert sequenced. The amplicons were sequenced in both senses by an external facility (Macrogen, Seoul, Korea) and deposited in the NCBI database under the accession numbers ON075837 to 41 for isolates Pc10, Pc280, Pc309, Pc190 and Pc60, respectively. The identity of the sequences was confirmed through BLAST analyses [42] at NCBI (http://www.ncbi.nlm.nih.gov/, accessed on 10 May 2022), searching nucleotide databases using a nucleotide query (BLASTn).

## Figures and Tables

**Figure 1 pathogens-11-00619-f001:**
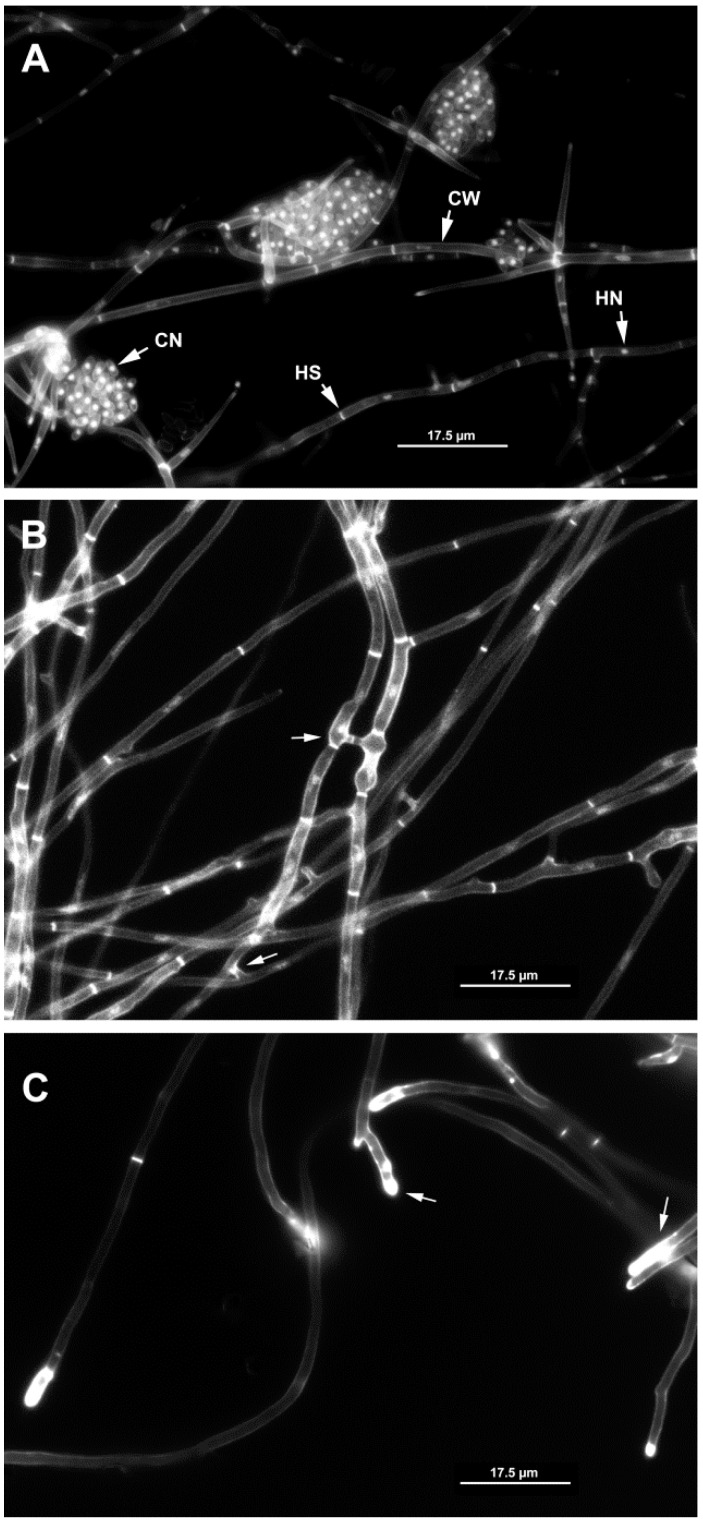
Fluorescence microscopy micrograph of hyphae stained with HOE 33258. (**A**): Pc309 cell walls (CW) of hyphae, conidium nucleus (CN) and hyphal nucleus (HN), hyphal septa (HS); (**B**): Pc309 mycelium showing different stages of hyphal anastomosis (arrows); (**C**): Pc190 hyphal tips and hyphal contact points (arrows). Scale bar: 17.5 μm.

**Figure 2 pathogens-11-00619-f002:**
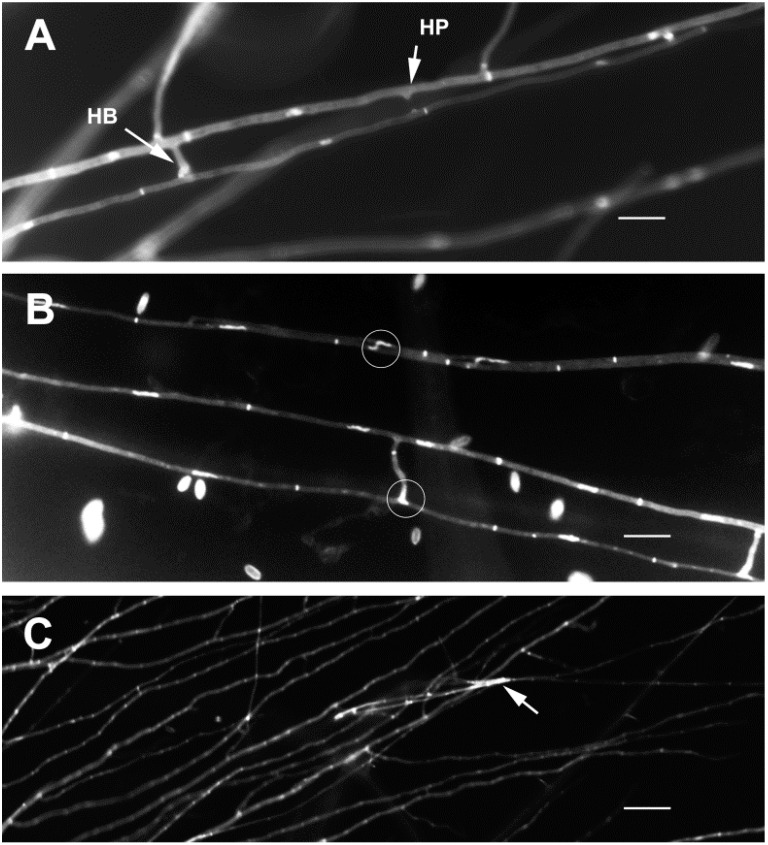
Fluorescence microscopy micrograph of anastomosing hyphae. (**A**): Pc309-Pc309 hyphal pegs (arrow) and bridges; (**B**): Pc60-Pc60 migrating nuclei through inconspicuous hyphal bridges (circled at the top) that were formed between two adjacent hyphae, fully developed hyphal bridge (circled at the bottom); (**C**): Pc280-Pc309 hyphal nuclei at hyphal contact point (arrow). HB = hyphal bridge, HP = hyphal peg. Scale bar (**A**,**B**): 5 μm, (**C**): 10 μm.

**Figure 3 pathogens-11-00619-f003:**
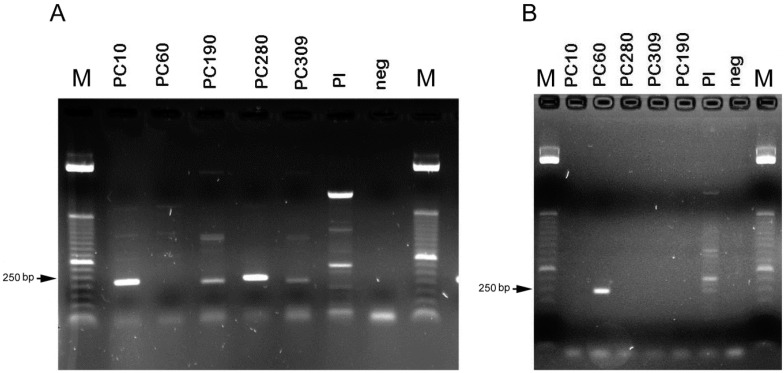
Agarose gels showing the mating-type amplification products in *Pochonia chlamydosporia* var. *chlamydosporia* isolates. (**A**): *MAT1-1-1*; (**B**): *MAT1**-2-1* primer combinations. No amplification product was obtained using other fungi or from the negative control (no template). M = marker (ladder) = 50 bp from Invitrogen, arrows indicate a 250 bp fragment. Pl = *Purpureocillium lilacinum*.

**Figure 4 pathogens-11-00619-f004:**
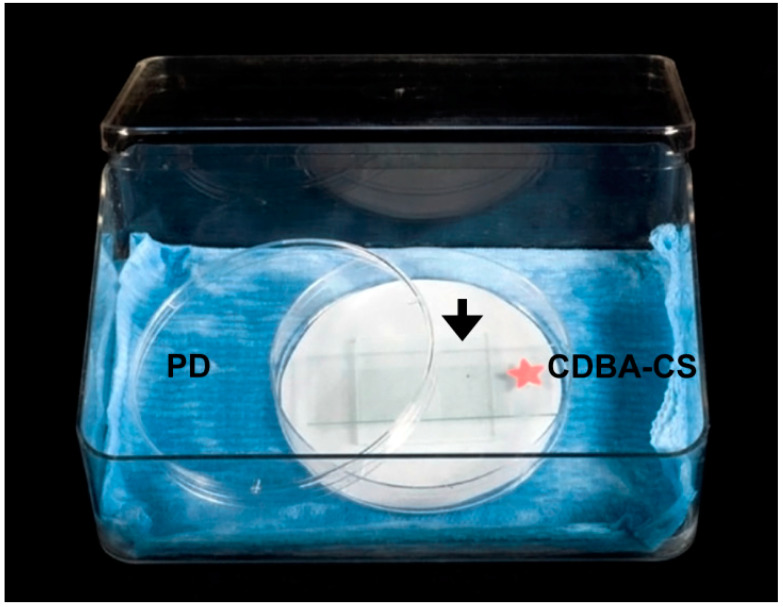
Humid chamber for anastomosis assay. Czapek Dox Broth Agar-coated glass slide (CDBA-CS) supported on a sterile glass square (arrow) placed in a Petri dish (PD) with moistened filter paper and placed inside a plastic box lined with moistened blue roll.

**Figure 5 pathogens-11-00619-f005:**
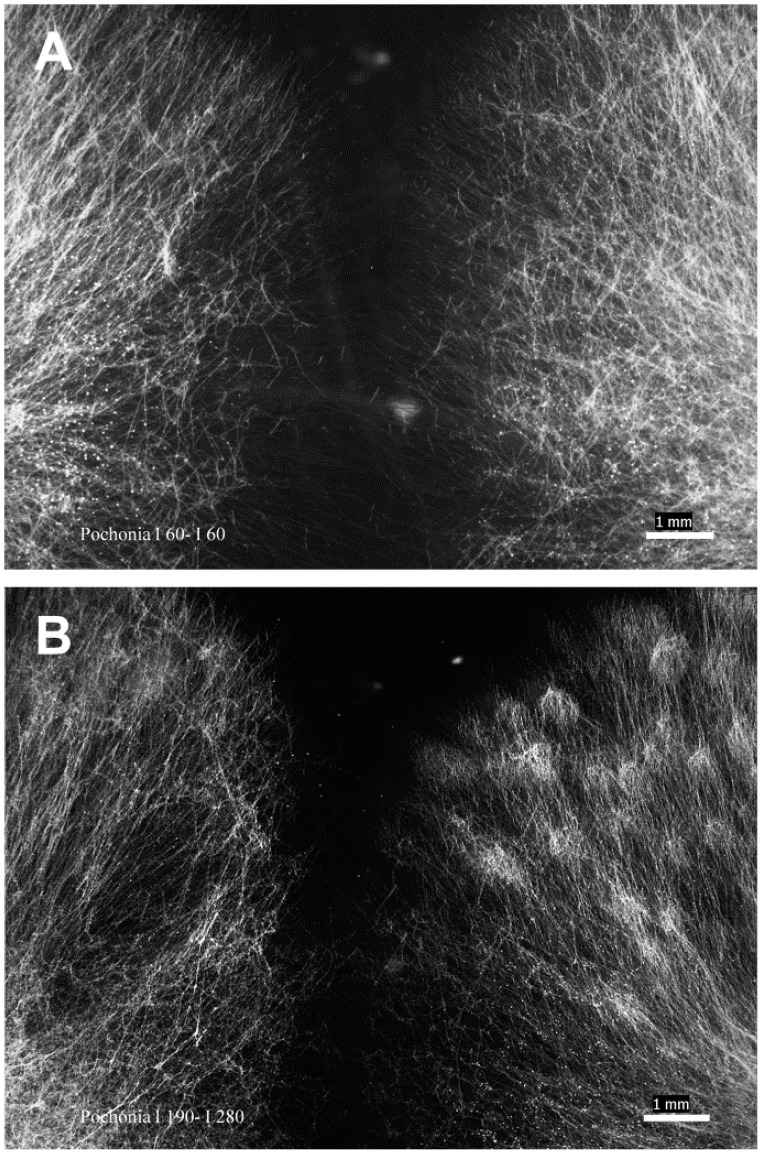
Light microscopy micrographs of isolates’ mycelia three days after culturing onto CDBA-CS for the in vitro anastomosis assay. Pairwise combinations (**A**): Pc60-Pc60 (1.25×), (**B**): Pc190-Pc280 (1.25×). CDBA-CS = Czapek Dox Broth Agar-coated glass slide. Scale bar: 1 mm (Wild M5 microscope).

**Table 1 pathogens-11-00619-t001:** Primers used for amplification of *Pochonia chlamydosporia* mating-type according to Yokoyama et al. [3].

Primer Name/Amplified Locus	Sequence (5′–3′) *
MAT1-F1 (*MAT1-1-1*)	CG(A/G)GC(A/T)AA(A/G)CG(A/G)CCATT(G/T)AA(C/T)GC
MAT1-R1 (*MAT1-1-1*)	TT(G/T)CCCATCTC(A/G)TC(A/G)CGGA(C/T)(A/G)AA(A/G)GA
MAT1-F2 (*MAT1-1-1*)	CCAAGCCGGTATCAGTGAATGC
MAT1-R2 (*MAT1-1-1*)	CGACCTGTTGTCGAACAAAGGT
MAT2-F1 (*MAT1-2-1*)	GC(A/G)TATATTCT(A/G)TACCGCAG
MAT2-R1 (*MAT1-2-1*)	CGAGGTTGATA(T/C)TGATA(T/C)TG
MAT2-F2 (*MAT1-2-1*)	ACGCATATATT(T/C)TGTACCG(T/C)AA
MAT2-R2 (*MAT1-2-1*)	GAAGGCTTTCG(A/T)GGT(T/C)TGTAC

* Reproduced with permission from Elsevier [license 5315351410835]. ©2004 Federation of European Microbiological Societies.

**Table 2 pathogens-11-00619-t002:** MAT1 genotype of *Pochonia chlamydosporia* var. *chlamydosporia* isolates according to Yokoyama et al. [3] PCR-based assay.

Species	Isolate	Biotype	Mating-Type PCR-Based Assay		
			*MAT1* *-1-1*	*MAT1* *-2-1*	Genotype
*Pochonia chlamydosporia*	Pc10	*Meloidogyne*	ON075837 ^a^	NA	*MAT1-1*
*Pochonia chlamydosporia*	Pc280	*Globodera*	ON075838 ^a^	NA	*MAT1-1*
*Pochonia chlamydosporia*	Pc309	*Meloidogyne*	ON075839 ^a^	NA	*MAT1-1*
*Pochonia chlamydosporia*	Pc190	*Meloidogyne*	ON075840 ^a^	NA	*MAT1-1*
*Pochonia chlamydosporia*	Pc60	*Heterodera*	NA	ON075841 ^a^	*MAT1-2*
*Purpureocillium lilacinum*	NA	NA	NA	NA	NA

^a^ Amplified, NA = Not amplified, *MAT1*-1-1 and *MAT1*-2-1 = mating-type genes [3,22] and corresponding NCBI accession number.

**Table 3 pathogens-11-00619-t003:** List of *Pochonia chlamydosporia* var. *chlamydosporia* and *Purpureocillium lilacinum* isolates (geographical origin and nematode biotype) that were included in the anastomosis assay and the mating-type molecular assay.

Fungus Species	Isolate	Country	Biotype
*Pochonia chlamydosporia* var. *chlamydosporia*	Pc10 **	Brazil	*Meloidogyne*
*P. chlamydosporia* var. *chlamydosporia*	Pc60 *	UK	*Heterodera*
*P. chlamydosporia* var. *chlamydosporia*	Pc190 *	Kenya	*Meloidogyne*
*P. chlamydosporia* var. *chlamydosporia*	Pc280 *	Jersey	*Globodera*
*P. chlamydosporia* var. *chlamydosporia*	Pc309 *	Zimbabwe	*Meloidogyne*
*Purpureocillium lilacinum*	**	South Africa	Unknown

* Isolates that were included in the anastomosis assay and the mating-type molecular assay. ** Isolates that were only included as positive controls for the mating-type molecular assay.

## Data Availability

The data will be freely accessible in NCBI after the article is published.

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
