# Peer review of "Exploring Anastomosis of Hyphae and Mating-Type Compatibility of Pochonia chlamydosporia Isolates of the Meloidogyne, Heterodera and Globodera Biotypes"

_pathogens, 2022, doi:10.3390/pathogens11060619_

Round 1

Reviewer 1 Report

Comments and suggestions are given in the separate file the Reviw report.

Author Response

26 April 2022

To whom may it concern,

We would like to thank the valuable comments, queries, suggestions and corrections that were kindly made to us by the referees. We have addressed and answered all queries of referees and the review report of the paper as much as possible. Figures have been edited to include scale bars, and gels resolution. We also have edited Tables 1 and 2 and prepared a new Table 3, which includes the whole list of isolates. We also have included extra references where needed. We are included the edited version in Word as an attached file.

We consider that the paper has been improved thanks to your feedback and pointed out the aspects that needed further clarification.

Kind regards,

Mariella Finetti-Sialer and Rosa H. Manzanilla-López

Review report of the article: Exploring Anastomosis of Hyphae and Mating Type Compatibility of Pochonia chlamydosporia Isolates of the Meloidogyne, Heterodera and Globodera Biotypes

 Overall Recommendation

The paper is in principle acceptable after revision based on the reviewer’s comments.

The objectives of the present study, in a few words, were to explore genetic exchange among different biotypes of P. chlamydosporia var. chlamydosporia originating from root-knot and cyst nematodes from two points of view, through an in vitro anastomosis assay and through a molecular approach.

Specific comments 

  1. The line 13

An anastomosis assay was made in vitro to assess hyphae vegetative compatibility and mating-type genotype of four isolates of P. chlamydosporia var. chlamydoporia of the Meloidogyne sp. (Pc10, Pc190, Pc309), Globodera sp. (Pc280) and Heterodera avenae (Pc60) biotypes, including 16 pairwise isolates combinations in four replicates.

Reply = four has been changed to five isolates.

  1. Keywords: Pochonia biotypes; MAT1-1-1; MAT1-2-1; biological control agents

R = It has been added the keyword anastomosis. The authors would like to keep also the key word biological control agents as it is related to Pochonia as one biological control agent.

  1. The lines 39-41:

The fungus is a facultative parasite of eggs and sedentary females of root-knot (Meloidogyne spp.), false root-knot (Nacobbus spp.,) and cyst nematodes (Globodera spp., Heterodera spp.). In addition, endo-

Comment (C) = The sentence requires references

Reply (R)= The classic and seminal references that confirmed the fungus parasitism of egg and females of most important nematode hosts have been added.

(Meloidogyne spp.) [8], false root-knot (Nacobbus spp.) [9] and cyst nematodes (Globodera spp., Heterodera spp.) [10]. In addition…

  1. Morgan-Jones, G.; White J.F.; Rodriguez-Kabana. R. Phytonematode pathology: Ultrastructural studies. I. Parasitism of Meloidogyne arenaria eggs by Verticillium chlamydosporium. Nematropica 1983, 13, 245–260
  2. Flores-Camacho, R.; Atkins, S.D.; Manzanilla-López, R.H.; Cid del Prado-Vera, I.; Martínez Garza, A. Isolation and PCR of five Mexican strains of Pochonia chlamydosporia var. chlamydosporia, potential biological control agent of Nacobbus aberrans sensu lato. Rev. Mex. Fitopatol. 2008, 26, 93–104.
  3. Kerry, B.R.; Crump, D.H. Observations on fungal parasites of females and eggs of the cereal cyst-nematode, Heterodera avenae, and other cyst nematodes. Nematologica 1977, 23,193–201, https://doi.org/10.1163/187529277x00543
  4. The lines 87-91:

Samples were screened for anastomosing hyphae with an inverted Olympus microscope using a 40× objective once every 24h for a period of five days. Selected samples were further assessed for migrating nuclei under fluorescence microscopy after isolate mycelia were stained with fluorescent HOE33258 dye [19].

R = We agree with the referee. The paragraph has been deleted as the information included in the section of materials and methods.

  1. The line 107:

Figure 1. Hyphae stained with HOE 33258. A: Pc309 cell walls (CW) of hyphae, conidium nucleus.

R= Information has been added as follows:

Figure 1. Fluorescence microscopy micrograph of hyphae stained with HOE 33258. A: Pc309 cell walls (CW) of hyphae,…

  1. The line 112:

Figure 2. Anastomosing hyphae. A: Pc309-Pc309 hyphal pegs (arrow) and bridges, B: Pc60-Pc60 mi

R= the information has been added as follows:

Figure 2. Fluorescence microscopy micrograph of anastomosing hyphae. A: Pc309-Pc309…

  1. The line 128:

Table 2. Sequence identity and MAT1 genotype of four isolates of Pochonia chlamydosporia var. chlamydosporia according to Yokoyama et al. [3].

C = Newly obtained sequences should be added in the table.

R = Table 2 has been revised and edited. There was confusion with sequence identity and sequence accession numbers.

  1. Figure 3

 R = The images of the gels have been improved as much as possible. An edited figure has been made and included in the revised version of the manuscript. The original two gels are slightly different in size as it has been pointed out and they are labeled now as A and B gels. The marker lane has been labeled as M.

  1. The line 132

 Agarose gels showing the mating type amplification products in Pochonia chlamydosporia var. chlamydosporia isolates obtained with the MAT 1-1-1 and MAT 1-2-1 primer combinations. No

 R = The A and B labels have been added to gels.

  1. The line 228

The four Pochonia isolates, plus one isolate of Purpureocillium lilacinum (Clavicipitaceae),

R = It has been corrected. Paecilomyces lilacinus (= Purpureocillium lilacinum) was previously included as a member of the Clavicipitaceae, i.e. Clade A of Sung et al. (2007). The family where P. lilacinum is included at present is Ophiocordicypitaceae, according to Index Fungorum (http://www.indexfungorum.org/names/names.asp). We have included in the same sentence the name of the family Clavicipitaceae to which five of the isolates belong. This way, the information is not repeated on line 298 (comment 14).

Sung, G-H; Hywel-Jones, N.L.; Sung, J-M., Luangsa-ard, J.J.; Shrestha, B.; Spatafora, J.W. Phylogenetic classification of Cordyceps and the clavicipitaceous fungi. Studies in Mycology 2007, 57, 5-59. https://pubmed.ncbi.nlm.nih.gov/18490993/

  1. The lines 233-237

All laboratory assays were made in a flow cabinet under sterile conditions. Monosporic culture of isolates was produced either through conidia dilutions [31, 32] or conidia picked individually with a Singer stage-mounted micromanipulator [33]. Monosporic cultures were subsequently cultured on fresh CMA and PDA media with antibiotics [34].

R= We would prefer to keep this section as it stands because section 4.1 refers only to the origin of the isolates, and following section 4.2 refers to how monosporic cultures were prepared from isolates samples and best laboratory practices followed i.e. sterile conditions used for the study. We instead have merged sections 4.2 and 4.3.

  1. The lines 261-264

Figure 4. Humid chamber for anastomosis assay. Czapek Dox Broth Agar-coated glass slide (CDBA-

R= We would prefer to keep the figure as it illustrates graphically the procedure of how the humid chamber was prepared to keep inside the CDBA coated slides. The reason that the CBBA coat is not noticed on the glass slide is because of the clear colour of the medium.

  1. The line 278

Figure 5. Isolate mycelia cultured three days after culturing onto CDBA-CS for the in vitro anasto-

R= The information has been included and it is now written as:

Figure 5. Ligth microscopy micrograph of isolates mycelia three days after culturing onto CDBA-CS for the in vitro anastomosis assay. Pairwise combinations A: Pc60-Pc60 (1.25×); B: Pc190-Pc280 (1.25×). CDBA-CS = Czapek Dox Broth Agar-coated glass slide. Scale bar: 1 mm (Wild M5 microscope).

  1. The line 298

Pochonia chlamydosporia var. chlamydosporia isolate (Pc10) and Purpureocillium lilacinum (Clavicipitaceae) were used as positive and …

R = We have deleted Clavicipitaceae and the correct family Ophiocordicypitaceae is now mentioned on l. 228 (see reply to comment 10)

  1. The line 400 References
  2. Rodríguez-Guerra, R.; Ramírez-Rueda, M.T.; Martínez de la Vega, O.; Simpson, J. Variation in genotype, pathotype and anastomosis groups of Colletotrichum lindemuthianum isolates from Mexico. Plant Pathol. 2002, 52, 228–235, https://bsppjournals.onlinelibrary.wiley.com/doi/full/10.1046/j.1365-3059.2003.00808.x

R= The reference has been included.

Reviewer 2 Report

I am finding the article well prepared. 

I think that designation of P. chlamydosporia isolates used for study is little bit confusing; it is clear that the research was conducted with 4 isolates from different localities, but you are also mentioning 4 more isolates obtained from Rothamsted Research collection; these were evaluated as well?

Maybe it will be good to include table summarizing all isolates used in your work to make this important part of the article clear and easier for the reader.

1st paragraph of the Results should be included into Material and methods rather.

Author Response

To whom may it concern,

We would like to thank the valuable comments, queries, suggestions and corrections that were kindly made to us by the referees. We have addressed and answered all queries of referees and the review report of the paper as much as possible. Figures have been edited to include scale bars, and gels resolution. We also have edited Tables 1 and 2 and prepared a new Table 3, which includes the whole list of isolates. We also have included extra references where needed.

We consider that the paper has been improved thanks to your feedback and pointed out the aspects that needed further clarification.

Kind regards,

Mariella Finetti-Sialer and Rosa H. Manzanilla-López

Referee 2

I am finding the article well prepared. 

I think that designation of P. chlamydosporia isolates used for study is little bit confusing; it is clear that the research was conducted with 4 isolates from different localities, but you are also mentioning 4 more isolates obtained from Rothamsted Research collection; these were evaluated as well?

R = They were only four isolates from the Rothamsted collection that were used both in both the anastomosis assay and the molecular mating-type assay. We have added information in the manuscript:

The information was written on lines 222-223 as:

The four P. chlamydosporia var. chlamydosporia isolates included in the in vitro anastomosis study were selected considering…

Now it is written as:

The four P. chlamydosporia var. chlamydosporia isolates included in the in vitro anastomosis and mating-type study were selected considering

Maybe it will be good to include table summarizing all isolates used in your work to make this important part of the article clear and easier for the reader.

R = We have prepared and included a new Table 3 for the isolates.

1st paragraph of the Results should be included into Material and methods rather.

R = The paragraph as been deleted.

Reviewer 3 Report

The article entitled: "Exploring Anastomosis of Hyphae and Mating Type Compatibility of Pochonia chlamydosporia Isolates of the Meloidogyne,
Heterodera and Globodera Biotypes" is intresting text concerning, in my opinion, useful and future-proof research. Although the subject is intresting I have some concerns about quality of results, which are listed below. The main worries refers to PCR results which now gives free space for interpretation.

Abstract: what was the aim of the studies? Hypothesis? This information must be added to the text.

Introduction

line 35:  "hyperparasite of fungi" - check this information, in citation 4, which I guess relates to those data, I did not found that information. 

line 36: This sentence is already written one sentence before. Try to link them.

line 45: " BCA vs a specific " use whole word

Results:

Figure 1: scale bar at figure B and C needed

line 122: I do not understand the nomenclature of the primers. Here you written MAT1-2-1 and I don't know with which primers in table it correspond 

line 123: delete " including primer sequences"

Figure 3: 

1. above you said that the product is 250 bp and here you give a hint (arrow) at 350 bp. It would be better to show the 250 bp size at the ladder

2. at this figure we can see additional, probably unspecific, products at the gel with bigger size than 250 bp (even bigger that 350 bp), it should not be this way.

Author Response

26 April 2022

To whom may it concern,

We would like to thank the valuable comments, queries, suggestions and corrections that were kindly made to us by the referees. We have addressed and answered all queries of referees and the review report of the paper as much as possible. Figures have been edited to include scale bars, and gels resolution. We also have edited Tables 1 and 2 and prepared a new Table 3, which includes the whole list of isolates. We also have included extra references where needed.

We consider that the paper has been improved thanks to your feedback and pointed out the aspects that needed further clarification.

Kind regards,

Mariella Finetti-Sialer and Rosa H. Manzanilla-López

Referee 3

The article entitled: "Exploring Anastomosis of Hyphae and Mating Type Compatibility of Pochonia chlamydosporia Isolates of the Meloidogyne, Heterodera and Globodera Biotypes" is interesting text concerning, in my opinion, useful and future-proof research. Although the subject is interesting I have some concerns about quality of results, which are listed below. The main worries refers to PCR results which now gives free space for interpretation.

Abstract: what was the aim of the studies? Hypothesis? This information must be added to the text.

R = Information has been added to the abstract.

Introduction

line 35:  "hyperparasite of fungi" - check this information, in citation 4, which I guess relates to those data, I did not found that information.

R= Reference 4 does not dealt with Pochonia hyperparasitism. We apologize for not having included a reference. The late Prof B. R. Kerry considered P. chlamydosporia as a mycoparasite of fungal oospores (pers. comm. to R. H. Manzanilla-López). Hyperparasitism has been reported in species of Verticillium closely related to Pochonia, which was formerly included within the genus Verticillium and Zare et al. (2001) refers to hyperparasitism of Verticillium epiphytium (Plectosphaerellaceae) a species parasite of rust fungi that was closely realted  to Pochonia (see reference below). 

Zare, R., Gams, W. and Evans, H.C. (2001). A revision of Verticillium section Prostrata. V. the genus Pochonia, with notes on Rotiferophthora. Nova Hedwigia 73: 51-86.

Therefore, we have added references and have modified the information in the manuscript as follows:

…an endophyte, an antagonist of some phytopatogenic fungi [4, 5], and a parasite of beetle larvae, mollusc eggs and PPN [6]. Pochonia chlamydosporia is also considered a successful BCA of PPN [7]…

There are other references related to the antagonism of V. chlamydosporium (= P. chlamydospora) to other plant pathogenic fungi. See references below. It is known that secondary metabolites of Verticilllium chlamydosporium (= Pochonia chlamydosporia) inhibited rusts (Leinhos and Buchenauer, 1992). 

  1. Leinhos, G.M.E.; Buchenauer, H. Inhibition of rust diseases of cereals by metabolic products of Verticillium chlamydosporium. J. Phytopathol. 1992, 136, 177–193.
  2. Monfort, E.; Lopez-Llorca, L.V.; Jansson, H.B.; Salinas, J.; Park, J.O.; Sivasithamparam, K. (2005). Colonisation of seminal roots of wheat and barley by egg-parasitic nematophagous fungi and their effects on Gaemannomyces graminis var. tritici & development of root-rot. Soil Biol. Biochem. 2005, 37, 1229–1235.
  3. Evans, H.C.; Kirk, P. Systematics of Pochonia. In Perspectives in Sustainable Nematode Management Through Pochonia chlamydosporia Applications for Root and Rhizosphere Health; Manzanilla-López, R.H.; Lopez-Llorca, L.V. Eds.; Springer, Cham, 2017; pp. 21–43, doi: 10.1007/978-3-319-59224-4_2.
  4. Siddiqui, I.A.; Atkins, S.D.; Kerry, B.R. Relationship between saprophytic growth in soil in different isolates of Pochonia chlamydosporia collected from cyst and root-knot nematodes and the infection of nematode eggs. Ann. Appl. Biol. 2009, 155, 131–141, https://onlinelibrary.wiley.com/doi/10.1111/j.1744-7348.2009.00328.x
  5. Morgan-Jones, G.; White J.F.; Rodriguez-Kabana. R. (1983) Phytonematode pathology: Ultrastructural studies. I. Parasitism of Meloidogyne arenaria eggs by Verticillium chlamydosporium. Nematropica 1983, 13, 245–260
  6. Flores-Camacho, R.; Atkins, S.D.; Manzanilla-López, R.H.; Cid del Prado-Vera, I.; Martínez Garza, A. Isolation and PCR of five Mexican strains of Pochonia chlamydosporia var. chlamydosporia, potential biological control agent of Nacobbus aberrans sensu lato. Rev. Mex. Fitopatol. 2008, 26, 93–104.
  7. Kerry, B.R.; Crump, D.H. Observations on fungal parasites of females and eggs of the cereal cyst-nematode, Heterodera avenae, and other cyst nematodes. Nematologica 1977, 23,193–201

line 36: This sentence is already written one sentence before. Try to link them.

  1. 33-35: Pochonia chlamydosporia var. chlamydosporia is characterized by diversity of econutritional groups [2, 3]. This fungus shows a variety of ecotrophic interactions, acting as a soil saprotroph, an endophyte, a hyperparasite of fungi, and a parasite of beetle larvae, mollusc eggs and PPN…

R = The sentence has been changed to:

Pochonia chlamydosporia var. chlamydosporia is characterized by diversity of econutritional groups [2, 3], acting as a soil saprotroph, an endophyte,..

line 45: " BCA vs a specific " use whole word

R = The BCA acronym meaning was written on l. 31-32:

…application of PPN biological control agents (BCA). Among…

However, it has been changed as requested to:

…application of a single biotype of the fungus as a biological control agent vs a specific host nematode…

Results:

Figure 1: scale bar at figure B and C needed

R = Scale bar has been included

line 122: I do not understand the nomenclature of the primers. Here you written MAT1-2-1 and I don't know with which primers in table it correspond 

R = Yokoyama refers to different set of primers, MAT1-F1 and MAT1-R1 that amplify MAT1-1-1 gene of the clavicipitaceous genera, while MAT1-F2 and MAT1-R2 are for Claviceps purpurea. MAT2 is determined by the mating-type locus MAT1-2-1 amplified with another set of primers. MAT2-F1 and MAT2-R1 for the clavicipitaceous genera and MAT2-F2 and MAT2-R2 for the clavicipitaceous genera Balansia (Ephelis), Claviceps and Epichloë (Neotyphodium), and anamorphs Engyodontium album, P. lilacinus (= Purpureocillium lilacinum) and P. nostocoides. The table was amended with the respective amplified locus

R = Information has been added to clarify the primers nomenclature (p. 2, l. 76-81)

The PCR-based mating-type assay developed by Yokoyama et al. [3, 23, 24] for Clavicipitaceae species provides a fast and practical tool to identify the mating-type genotype of P. chlamydosporia strains or biotypes, without the need for mating experiments.  Yokoyama et al. [3] designed primers sets for identifying the mating-type genes MAT1-1-1 and MAT1-2-1 based on the aminoacid sequences of the conserved alpha and HMG boxes [3, 22]. Furthermore,

line 123: delete " including primer sequences"

R= the sentence has been deleted

Figure 3: 

  1. above you said that the product is 250 bp and here you give a hint (arrow) at 350 bp. It would be better to show the 250 bp size at the ladder.

R = We have done the change as requested and Figure 3 has been edited accordingly.

  1. at this figure we can see additional, probably unspecific, products at the gel with bigger size than 250 bp (even bigger that 350 bp), it should not be this way.

R = Working with total DNA, according with the PCR kinetics, some spurious bands could appear, but are quite distinguishable from the target.

Round 2

Reviewer 3 Report

I accept the responses to the review, except for the point which is quite crucial for the whole article, namely the PCR reaction.

I agree that when working with total DNA, there may be some non-specific bands, for example related to pseudogens. It would be better to consider working on mRNA in order to obtain a more specific product, and certainly when the results is such a high non-specific it is necessary to check the sequence of the resulting product using sequencing Sanger. Did you do that? That could solve the problem.

It's not about guessing the result - the result should be obvious. In this way, it will be possible to obtain information whether the obtained products are what you expected. I believe that optimizing PCR reactions to achieve high specificity of reactions is a normal activity in molecular biology.

Author Response

Reviewer 3

Comments and Suggestions for Authors

I accept the responses to the review, except for the point which is quite crucial for the whole article, namely the PCR reaction.

I agree that when working with total DNA, there may be some non-specific bands, for example related to pseudogens. It would be better to consider working on mRNA in order to obtain a more specific product, and certainly when the results is such a high non-specific it is necessary to check the sequence of the resulting product using sequencing Sanger. Did you do that? That could solve the problem.

Reply from authors: Yes indeed, working with the messenger RNA would give us an Electrophoretic profile without non-specific bands, that is why we picked out bands of the expected dimension, cloned in pGemT vector and sequencing it (with the Sanger protocol done by an external service: Macrogen), described in M&M, page 13, lines 327 and 328. For any future assessment it would be recommended to work with the mRNA.

It's not about guessing the result - the result should be obvious. In this way, it will be possible to obtain information whether the obtained products are what you expected. I believe that optimizing PCR reactions to achieve high specificity of reactions is a normal activity in molecular biology.